# Functional Characterization of the *MeSSIII-1* Gene and Its Promoter from Cassava

**DOI:** 10.3390/ijms25094711

**Published:** 2024-04-26

**Authors:** Xiao-Hua Lu, Ya-Jie Wang, Xing-Hou Zhen, Hui Yu, Mu Pan, Dong-Qing Fu, Rui-Mei Li, Jiao Liu, Hai-Yan Luo, Xin-Wen Hu, Yuan Yao, Jian-Chun Guo

**Affiliations:** 1National Key Laboratory for Tropical Crop Breeding, School of Life and Health Sciences, Hainan University, Haikou 570228, China; 19071010110006@hainanu.edu.cn (X.-H.L.); 2021110710000044@hainanu.edu.cn (X.-H.Z.); 21220860000016@hainanu.edu.cn (M.P.); huxinwen@hainanu.edu.cn (X.-W.H.); 2National Key Laboratory for Tropical Crop Breeding, Sanya Research Institute, Institute of Tropical Bioscience and Biotechnology, Chinese Academy of Tropical Agricultural Sciences, Haikou 571101, China; wangyajie@itbb.org.cn (Y.-J.W.); 18417094705@163.com (H.Y.); liruimei@itbb.org.cn (R.-M.L.); liujiao@itbb.org.cn (J.L.); 3School of Tropical Agriculture and Forestry, Hainan University, Haikou 570228, China; 22210901000047@hainanu.edu.cn; 4Tropical Crops Genetic Resources Institute, Chinese Academy of Tropical Agricultural Sciences, Haikou 571101, China; haiyan2022@catas.cn

**Keywords:** cassava, starch synthase, *MeSSIII-1*, promoter, hormones, starch synthesis

## Abstract

Soluble starch synthases (SSs) play important roles in the synthesis of cassava starch. However, the expression characteristics of the cassava SSs genes have not been elucidated. In this study, the *MeSSIII-1* gene and its promoter, from SC8 cassava cultivars, were respectively isolated by PCR amplification. MeSSIII-1 protein was localized to the chloroplasts. qRT-PCR analysis revealed that the *MeSSIII-1* gene was expressed in almost all tissues tested, and the expression in mature leaves was 18.9 times more than that in tuber roots. *MeSSIII-1* expression was induced by methyljasmonate (MeJA), abscisic acid (ABA), and ethylene (ET) hormones in cassava. *MeSSIII-1* expression patterns were further confirmed in *proMeSSIII-1* transgenic cassava. The promoter deletion analysis showed that the −264 bp to −1 bp *MeSSIII-1* promoter has basal activity. The range from −1228 bp to −987 bp and −488 bp to −264 bp significantly enhance promoter activity. The regions from −987 bp to −747 bp and −747 bp to −488 bp have repressive activity. These findings will provide an important reference for research on the potential function and transcriptional regulation mechanisms of the *MeSSIII-1* gene and for further in-depth exploration of the regulatory network of its internal functional elements.

## 1. Introduction

Gene expression is usually regulated at the transcriptional and post-translational levels [1]. Regulation at the transcriptional level is the most critical link between gene expression and multistage regulation, in which promoters play an important role at the transcriptional level [2]. Multiple *cis*-acting elements are distributed on promoter sequences, including response elements that are responsive to light, abiotic stresses, hormones, and other stimuli, which may be binding sites for transcription factors [3]. Generally, the function of *cis*-acting elements can be resolved by truncations of the promoter sequence at the 5’ end. In plants, recombinant promoter::GUS (β-glucuronidase gene) constructs and *Agrobacterium*-mediated transient expression systems, as well as transgenic promoter plants, are commonly used to analyze promoter activity [4]. In plant genetic engineering, promoters are commonly used to regulate gene expression in order to improve crop quality, increase crop yield, and enhance crop adaptability under adverse conditions [5]. Starch synthesis is a complex metabolic process, and most genes involved in starch synthesis in plants have been identified and isolated. However, little is known about how expression of these key genes is regulated by promoters. Promoters, as important regulatory elements of gene expression, play a crucial role in starch synthesis. Editing of *Wx* allele promoters in rice using CRISPR/Cas9 technology has reduced amylose content and improved the quality of rice grains [6,7].

Starch, one of the major sources of calories in human diet, is synthesized in most vascular plants [8]. In most higher plants, there are two main types of starch, storage starch and transitory starch, based on biological function. Storage starch is produced in amyloplasts for long-term energy storage. Transient starch is synthesized and degraded in chloroplasts within photosynthetic tissues according to the diurnal cycle [9,10]. Starch exists mainly as two polymers: amylose and amylopectin. Amylose is a linear polymer linked by α-1,4-glycosidic bonds, and accounts for 10% to 35% of starches in plants. While amylopectin is a highly branched glucan linked by α-1,6-glucosidic bonds, accounting for 75% to 90% of most plant starches [11]. The process of starch biosynthesis is the result of the synergistic action of multiple starch synthesis-related enzymes (Figure 1). The ADP-glucose pyrophosphorylase (AGPase, EC:2.7.7.27) catalyzes glucose-1-phosphate (Glc-1-P) and ATP to produce the activated glucosyl donor ADP glucose [12]. Granule-bound starch synthase (GBSS, EC 2.4.1.242) is primarily involved in amylose production. Soluble starch synthase (SSs, EC.4.1.21), starch branching enzyme (BE, EC 2.4.1.18), and starch debranching enzymes (Isoamylase, ISA, EC 3.2.1.68; Pullulanase, PUL, EC 3.2.1.41) act synergistically in the process of amylopectin synthesis [13,14].

Cassava (*Manihot esculenta* Crantz) is a major food crop in tropics, with a very high starch content in its tuber roots. In cassava, the starch content of fresh tuber roots is up to 32%, of which amylose accounts for 10–35% and amylopectin accounts for 75–90% [15]. The ratio of amylose/amylopectin affects food quality and industrial application of starches. Starches with high amylose contents are highly resistant starches, which can reduce the risk of obesity, diabetes, and cardiovascular disease in patients [16,17]; high amylose is also used in biodegradable plastics and other industries [18,19]. High amylopectin starches can be used as stabilizers and thickeners in foods and industrial alcohols due to their unique functional properties [20]. 

There are nine starch synthase genes in the cassava genome, including two GBSSs and seven soluble starch synthase genes [21]. GBSSs play key roles in the elongation of amylose by catalyzing the addition of ADPG to the nonreducing end of the glucan chain. RNAi technology was utilized to downregulate MeGBSSI activity in cassava, resulting in a decrease ranging from 2% to 21% in the amylose content of transgenic plants [22]. Targeted mutation of *MeGBSSI* using CRISPR/Cas9 gene editing produces waxy starches without amyloses in cassava tuber roots [23]. Seven encoded soluble starch synthase enzymes are named as MeSSI, MeSSII-1, MeSSII-2, MeSSIII-1, MeSSIII-2, MeSSIV, and MeSSV in cassava [21]. Downregulation of MeSSII-1 expression in cassava led to a significant reduction in total starch content and changes in starch chain length distribution [24]. Mutation of OsSSIIIa in rice led to an increase in amylose content and changes in chain length distribution, indicating that OsSSIIIa plays an important role in starch synthesis by prolonging long amylopectin chains [25].

Hormones play a crucial role in all stages of plant growth and development, including starch synthesis. Treatment of cassava leaves with the exogenous hormone ABA during tuber root formation revealed that starch synthesis-related genes were regulated and affected the rate of temporary starch synthesis in the leaves [26]. A positive correlation was found between ABA content and starch synthesis-related enzyme activities in stems, leaves, and tuber roots, suggesting that endogenous ABA in cassava may affect the accumulation of tuber starch by regulating key enzymes [27]. However, there is no direct evidence that hormones regulate expression of genes related to starch synthesis and affect starch synthesis. Among seven soluble starch synthesis genes in cassava, the *MeSSIII-1* gene is highly expressed in different tissues or organs and during the development of root tubers (unpublished SC8 cassava RNA sequencing data). In this study, we isolated the *MeSSIII-1* gene and promoter from SC8 cassava cultivars, and the expressing characterizations of the *MeSSIII-1* promoter have been analyzed.

## 2. Results

### 2.1. Isolation and Characterization of MeSSIII-1

Based on the NCBI database, the full-length cDNA sequence of *MeSSIII-1* was successfully cloned from an SC8 cassava cultivar. The CDS sequence of *MeSSIII-1* is 3441 bp in length, which encodes 1146 amino acids, and is located on chromosome 16 (Figure 2A). The molecular weight and theoretical isoelectric point of *MeSSIII-1* are 130.62 kDa and 6.66, respectively. The exon–intron analysis indicated that *MeSSIII-1* has 16 exons and 15 introns, with the second exon being the smallest and the third exon being the largest (Figure 2B). The evolutionary relationship between MeSSIII-1 and SSIIIs in other species indicates that its closest protein homologue is HbSSIII from *Hevea brasiliensis*, and it is also closer to RcSSIII from *Ricinus communis* and PtSSIII from *Populus trichocarpa*, while it is more distantly related to OsSSIII-1 of the monocotyledonous plant of the *Oryza sativa* Japonica Group (Figure 2C).

### 2.2. Subcellular Localization of MeSSIII-1 Protein

To investigate the localization of the MeSSIII-1 protein, a pCAMBIA1300-35S-MeSSIII-1::GFP fusion expression vector was constructed and transiently expressed in tobacco epidermal cells. The empty vector pCAMBIA1300-GFP was used as a control. The results showed that the control GFP fluorescence was observed in the cytoplasm and nucleus, whereas the fluorescence signal of MeSSIII-1::GFP was observed only in chloroplasts, indicating that the MeSSIII-1 protein was localized to chloroplasts (Figure 3).

### 2.3. Expression Patterns of MeSSIII-1 in Cassava Tissues 

The expression of the *MeSSIII-1* gene in cassava tissues or organs was analyzed by qRT-PCR. The results showed that the expression of the *MeSSIII-1* gene was highly expressed in mature leaves and axillary buds, and the expression in mature leaves was 18.9 times more than that in tuber roots (Figure 4).

### 2.4. Isolation and Cis-Element Analysis of MeSSIII-1 Promoter

A 1228 bp promoter sequence, upstream of the ATG start codon of the *MeSSIII-1* gene, was cloned, *proMeSSIII-1*. The *cis*-acting elements of *proMeSSIII-1* were analyzed. It was found that *proMeSSIII-1* contained basic promoter elements, such as TATA and CAAT boxes (Appendix A and Appendix A ); abscisic acid (ABA) response element (ABRE); MeJA response elements CGTCA-motif and TGACG-motif; and ethylene response elements (ERE) (Table 1). These findings suggest that *proMeSSIII-1* may be involved in hormone signaling responses.

### 2.5. Expression Profiles of MeSSIII-1 Gene under Hormone Treatments

To investigate the effects of exogenous hormones on *MeSSIII-1* gene expression, the expression patterns of *MeSSIII-1* in cassava histocultured seedlings and tuber roots at the expansion stage under hormones of MeJA, ABA, and ET treatments were analyzed by qRT-PCR. In cassava histocultured seedlings, for MeJA treatment, the expression of *MeSSIII-1* in leaves and roots was significantly downregulated at 2 h; in stems, it also decreased at 2 h and was significantly upregulated at 24 h. The induced expressions were more sensitive in stems than in leaves and roots, reaching a peak (2.4-fold) at 24 h in stems (Figure 5A). For ABA treatment, the induced expressions of the *MeSSIII-1* gene were more sensitive in roots then in leaves, with less induced expression in stems; they reached a peak (6-fold) at 24 h in roots, and reached a peak (4.44-fold) at 12 h in leaves, while in stems, the increased expression was only 2.22-fold of 0 h at 24 h (Figure 5B). For ET treatment, the induced expressions of the *MeSSIII-1* gene were more sensitive in leaves, lesser in stems and roots; they reached a peak (13-fold) at 24 h in leaves, while in stems and roots, the increased expression was only 4.69-fold of 0 h in stems at 12 h, and 4.76-fold of 0 h in roots at 24 h (Figure 5C).

In tuber roots, the expressions of the *MeSSIII-1* gene were induced by hormones of MeJA, ABA, and ET. For MeJA treatment, the expression of the *MeSSIII-1* gene reached a peak (1.92-fold) at 2 h. For ABA treatment, the expression of the *MeSSIII-1* gene reached a peak (3.33-fold) at 12 h. For ET treatment, the expressions of the *MeSSIII-1* gene were increased at 12 h and 24 h, and reached a peak (2.87-fold) at 24 h (Figure 5D).

### 2.6. Analysis of MeSSIII-1 Promoter Activity 

Based on the distribution of cis-acting elements, *proMeSSIII-1* was truncated into four segments: −987 bp (SP1), −747 bp (SP2), −488 bp (SP3), and −264 bp (SP4) (Figure 6B). The 35S promoter in the pCAMBIA1304-35S-GUS::GFP (*GUS*, *β-glucuronidase*) vector was replaced with each of four truncated fragments and the *proMeSSIII-1* to drive GUS expression (Figure 6A). To detect activity of the five different length promoters, GUS staining and GUS activity were performed on tobacco leaves treated with *Agrobacterium*-mediated transient transformation. The GUS activity was the strongest in the transformation of *proMeSSIII-1*, followed by 35S and SP3. The lowest transformation was in SP4, which in order of *proMeSSIII-1* > 35S > SP3 > SP1 > SP2 > SP4 (Figure 6C,D). These results indicated that the region from −264 bp to −1 bp is core promoter. The regions from −987 bp to −1228 bp and from −488 bp to −264 bp significantly improve promoter activity. The regions from −987 bp to −747 bp and −747 bp to −488 bp have repressive domains.

### 2.7. Histochemical Localization of proMeSSIII-1 in Cassava

To reveal activity of the *MeSSIII-1* promoter in cassava, the pCAMBIA1304-*proMeSSIII-1*-GUS::GFP vector was transformed into SC8 cassava (Appendix A). The *proMeSSIII-1* transgenic cassava was identified by PCR amplification of the *gusA* gene (Appendix A). It was found that *proMeSSIII-1* transgenic cassava had strong GUS staining (Figure 7A). To investigate tissue-specific expression patterns of the *MeSSIII-1* promoter, the *proMeSSIII-1* transgenic cassava was used for histochemical GUS staining and GUS activity assay, including leaves, petioles, stems, fibrous roots, and tuber roots. The results showed that the leaves were strongly stained, followed by stems and tuber roots, the petioles and fibrous roots had less GUS staining (Figure 7B). In a GUS activity assay, the results showed that *proMeSSIII-1* is active in leaves, stems, and tuber roots. *proMeSSIII-1* has highest activity in leaves, followed by in stems and tuber roots, and the lowest activity was in fibrous roots (Figure 7C). Let us hypothesize that the *MeSSIII-1* promoter plays an important role in cassava leaves, stems, and tuber roots.

### 2.8. Activities of MeSSIII-1 Promoter in Response to MeJA, ABA, and ET Hormones 

To explore the role of the *proMeSSIII-1* in response to MeJA, ABA, and ET hormones, the whole *proMeSSIII-1* transgenic cassava histocultured seedlings were treated with MeJA, ABA, and ET, respectively. The results showed that activity of the *MeSSIII-1* promoter was induced by MeJA, ABA, and ET hormones (Figure 8A). Under ABA treatment, *proMeSSIII-1* activity reached a peak at 12 h, whereas it reached a peak at 24 h under MeJA and ET treatments. The peak activity increase was 1.63-fold in MeJA, 2.41-fold in ABA, and 2.72-fold in ET compared to 0 h treatment (Figure 8B).

To evaluate the effect of hormones on cassava tuber roots, the tuber root sections of *proMeSSIII-1* transgenic cassava were treated with MeJA, ABA, and ET. Promoter activity was analyzed by GUS staining and GUS activity. For MeJA treatment, GUS activity reached a peak (1.24-fold) at 24 h. For ABA treatment, GUS activity reached a peak (1.66-fold) at 6 h. For the ET treatment, GUS activity reached a peak at 12 h (1.57-fold). The GUS staining and GUS activity showed similar results (Figure 9).

## 3. Discussion

In plants, starch synthase SSIIIs play important roles in starch synthesis and are mainly involved in regulating the interactions between components in the amylopectin assembly system [28]. In different species, the number of SSIII genes varies and generally consists of 1-2 homologous genes. For example, there is one *AtSS3* gene in arabidopsis that has the highest expression in leaves, and the *AtSS3* mutation affects the synthesis of temporary starch in leaves [29]. *OsSSIII* has two isoforms in rice, *OsSSIIIa* and *OsSSIIIb*; *OsSSIIIb* is mainly expressed in leaves, while *OsSSIIIa* is expressed in endosperm [30]. It was found that *OsSSIIIb*-deficient mutants in rice showed a significant decrease in total starch content of leaves, while *OsSSIIIa*-deficient mutants showed no significant change in starch content, but with altered amylopectin structure, amylose content, and starch granules in endosperms [31,32]. Two isoforms, *MeSSIII-1* and *MeSSIII-2*, have been identified in cassava, and the expression of *MeSSIII-1* was higher than *MeSSIII-2* in different tissues (unpublished transcriptome data from our laboratory). In our study, it was found that *MeSSIII-1* was most highly expressed in mature leaves (Figure 4). *proMeSSIII-1* transgenic cassava detection revealed that the activity of *MeSSIII-1* promoter mainly in leaves (Figure 7). Cassava leaves are the primary site for temporary starch synthesis [33]. Genes play different roles due to their expression characteristics in different tissues. Therefore, we hypothesized that *MeSSIII-1* plays an important role in starch synthesis within the source organs, especially in the synthesis of temporary starch in the leaves.

The subcellular localization of genes is closely related to their functions. Our study revealed that the MeSSIII-1 protein is localized in chloroplasts. The potato IbSSI protein is located on chloroplasts, where starch biosynthesis occurs [34]. Previous studies have reported that arabidopsis SSIV interacts with plastid-related proteins through specific localization on the thylakoid membrane of chloroplasts, participating in the production of starch granules and controlling the number of starch granules in each chloroplast [35,36]. Transient starches are synthesized in cassava leaf chloroplasts during daytime and degraded at night-time, regulating the transport of photosynthetic assimilates from source to sink [37]. Therefore, we hypothesized that the *MeSSIII-1* protein is mainly involved in the synthesis of temporary starch in chloroplasts.

Transcriptional regulation plays an important role in gene activation or repression [38]. A promoter is an important transcriptional regulator in the regulation of gene expression [39,40]. *Cis*-acting elements in promoters play a key role in regulating gene expression at the transcriptional level [41,42]. To investigate the core region of the *MeSSIII-1* promoter, the *MeSSIII-1* promoter is truncated. GUS activity in transient tobacco leaves showed that the −264 bp to −1 bp *MeSSIII-1* promoter had basal activity. The region from −1228 bp to −987 bp and −488 bp to −264 bp significantly enhanced promoter activity. The regions from −987 bp to −747 bp and −747 bp to −488 bp might have repressive activity (Figure 6). These findings will provide useful information for regulating the accumulation of cassava starch by editing regulatory regions of the *MeSSIII-1* promoter to finely control the activity of starch synthase. For example, knocking out the inhibitory region of the *MeSSIII-1* promoter, which may lead to an enhanced expression of *MeSSIII-1*, could potentially promote the synthesis of starch.

Phytohormones play crucial roles in various stages of plant growth and development, including the process of starch synthesis, while the regulatory relationship between phytohormones and genes related to starch synthesis has been studied less. In this study, the prediction of *MeSSIII-1* promoter *cis*-acting elements revealed the presence of three hormone elements, MeJA, ABA, and ET, on its sequence (Table 1). Hormone treatments indicated that the expression of *MeSSIII-1* gene is induced by MeJA, ABA, and ET. Hormone treatments of *proMeSSIII-1* transgenic cassava revealed that GUS activity is induced by MeJA, ABA, and ET. It was found that *ZmSSIIIa* in maize was mainly expressed in the endosperm and was induced by ABA, and the analysis of its promoter activity showed similar results; the activity of the *ZmSSIIIa* promoter was highly expressed in the endosperm and was an ABA-inducible promoter [43]. *ZmEREB156* positively regulates *ZmSSIIIa* through the synergistic action of sucrose and ABA [44]. ABA content in tuber roots showed a significantly positive correlation with the activities of synergistic adenosine diphosphate glucose pyrophosphorylase (AGPase), soluble starch synthase (SSs), and sucrose phosphate synthase (SPS) [27]. Exogenous ethyleneglycol repressed the expressions of most starch synthesis genes such as *SUS*, *AGPase*, and *SSs,* and downregulated their enzymatic activities [45]. The region from −453 bp to −388 bp of the *MeSSIIb* promoter was a repressive domain of ethylene [46]. Luo found that ethylene treatment promoted accumulation of the starch from tuber roots and increased tuber yield in cassava [47]. MeJA has been found to promote starch accumulation in several plant species including arabidopsis, tobacco, and spinach, due to the upregulation of starch synthetic genes’ expression, such as genes encoding APS1, APL4, GBSS1, SS2, and SS3 [48]. Therefore, it is speculated that MeJA, ABA, and ET influence the synthesis of cassava starch by regulating the expression of *MeSSIII-1*.

In conclusion, this study has elucidated the characteristics of the *MeSSIII-1* promoter, offering a novel approach to regulate the activity of SSIII by modulating the *MeSSIII-1* promoter. In future research, we will utilize gene editing technology to precisely insert more hormone-responsive elements into the *MeSSIII-1* promoter, in order to enhance the positive regulatory capacity of hormones on the expression of the *MeSSIII-1* gene. These efforts could potentially lead to an increase in the yield of cassava.

## 4. Materials and Methods

### 4.1. Plant Materials and Growth Conditions

South China 8 (SC8) cassava cultivar used in this study was planted in Lingao, Hainan, China. Tissues (terminal buds, axillary buds, petioles, young leaves, mature leaves, stems, fibrous roots, and tuber roots) of SC8 cassava grown for 180 d were collected and the samples were immediately frozen in liquid nitrogen, followed by storage at −80 °C until RNA isolation. The *proMeSSIII-1* transgenic cassava plants were planted in field in Haikou, Hainan Province, China. SC8 cassava histocultured seedlings were kept in MS medium (supplemented with 2% sucrose and 0.8% agarose, pH 5.8, 121 °C 20 min autoclave sterilization) at 28 °C under 16 h/8 h (light/darkness) for 30 d to obtain consistent cassava seedlings.

### 4.2. Bioinformatics Analysis of the MeSSIII-1 Gene and Promoter

The coding sequence (CDS) and 1228 bp upstream of the translational start site (ATG) promoter sequence information of *MeSSIII-1* gene were obtained in the NCBI database (https://www.ncbi.nlm.nih.gov/, accessed on 12 January 2021). The physicochemical properties of MeSSIII-1 protein were predicted using the ProParam database (https://web.expasy.org/protparam/, accessed on 10 March 2021). The gene structure of *MeSSIII-1* was analyzed using GSDS version 2.0 (http://gsds.cbi.pku.edu.cn/index.php, accessed on 22 March 2021). The phylogenetic tree was constructed by MEGA7.0 software based on the neighbor-joining (NJ) approach followed by 1000 bootstrap replicates. The PlantCARE database (http://bioinformatics.psb.ugent.be/webtools/plantcare/html/, accessed on April 2021) was used to predict *cis*-elements of the *MeSSIII-1* promoter (Appendix A).

### 4.3. Cloning of MeSSIII-1 Gene and Promoter of Cassava

Total RNAs were extracted using M5 HiPer Plant RNeasy Complex Mini Kit (Mei5bio, Beijing, China) according to the kit instructions. cDNAs were synthesized with a MonScript^TM^ RTIII All-in-One Mix with dsDNase (Monad) in a 20 μL reaction mixture containing 1 μg of total RNA according to the manufacturer’s instructions. The PCR program was as follows: 37 °C for 5 min, 55 °C for 30 min, and 85 °C for 5 min. PCR amplification was carried out using cDNA as the template and MeSSIII-1-F/R (Appendix A) as specific primers with PrimeSTAR HS (Premix) DNA polymerase (TaKaRa, Beijing,, China). The PCR program was as follows: 94 °C for 5 min, followed by 35 cycles of 98 °C for 10 s, 58 °C for 30 s, and 72 °C for 3.5 min, following a 10 min extension at 72 °C. PCR products were detected by 1% agarose gel electrophoresis. Then, the samples were sent to Sangon Biotech (Shanghai, China) for sequencing.

Total DNA was extracted using M5 HiPer Plant Genomic DNA Kit (Mei5bio, Beijing, China) according to the kit instructions. The MeSSIII-1 promoter was isolated using Tks Gflex^TM^DNA polymerase (TaKaRa). The specific primers proMeSSIII-1-F/R (Appendix A) were designed based on the sequence of proMeSSIII-1. The PCR program was as follows: 94 °C for 1 min, followed by 30 cycles of 98 °C for 10 s, 55 °C for 15 s, and 68 °C for 60 s. PCR products were detected by 1% agarose gel electrophoresis and recovered by PCR product recovery kit (Sangon Biotech, Shanghai, China). Then, the cloned segment was inserted into pCAMBIA1304-35S-GUS::GFP vector for validity sequencing.

### 4.4. Subcellular Localization of MeSSIII-1 Protein

The CDS of *MeSSIII-1* without stop codons was amplified by PCR with the 1300-MeSSIII-1-F/R primers containing *Kpn* I and *Sal* I restriction sites (Appendix A). The amplified product was recombined into pCAMBIA1300-35S-GFP by homologous recombination method to form the recombinant plasmid pCAMBIA1300-35S-MeSSIII-1::GFP. The fusion vector pCAMBIA1300-MeSSIII-1-GFP and empty vector pCAMBIA1300-35S-GFP(control) was transformed into epidermal cells from tobacco (Nicotiana benthamiana) in the 4–6 leaves stage, using the Agrobacterium tumefaciens strain LBA4404 [49]. They were incubated for 48 h in a greenhouse at 22 °C under a 16 h light/8 h dark cycle at 50% relative humidity. The GFP fluorescence (excitation 488 nm, emission 500–550 nm) and chlorophyll autofluorescence (excitation 640.6 nm, emission 663–738 nm) were observed under a confocal microscope (LEICA, TCP SP8, Wetzlar, Germany).

### 4.5. Expression Pattern of MeSSIII-1 Gene

The expression of *MeSSIII-1* gene in eight tissues (terminal buds, axillary buds, petioles, young leaves, mature leaves, stems, fibrous roots, and tuber roots) was analyzed by qRT-PCR. To analyze the transcriptional levels of *MeSSIII-1* in response to hormone treatment, 1-month-old cassava histocultured seedlings, and cassava tuber slices at the tuber expansion stage were treated with 100 μM MeJA, 100 μM ABA, or 100 μM ET. The leaves, stems, roots, and tuber slices collected at 0 h, 2 h, 6 h, 12 h, and 24 h after hormone treatments, and were used for total RNAs extraction. Quantitative PCR was performed with a MonScript^TM^ ChemoHS qPCR Mix (Monad) using qMeSSIII-1-F/R (Appendix A) as specific primers. The qRT-PCR procedure was performed using the following profile: preincubated at 95 °C for 10 min, followed by 40 cycles of denaturation at 95 °C for 10 s, annealed at 60 °C for 10 s, and extended at 72 °C for 30 s.

### 4.6. Construction of Truncated MeSSIII-1 Promoter Vectors and Tobacco Leaves Injection

To identify core region of the *MeSSIII-1* promoter, the 1228 bp promoter sequence was consecutively truncated into four segments by PCR based on the distribution of *cis*-acting elements and named as SP1, SP2, SP3, and SP4. The four fragments and the full-length 1228 bp sequence were substituted for the 35S promoter within the pCAMBIA1304-35S-GUS::GFP vector by homologous recombination method. The five recombinant vectors and the original pCAMBIA1304-35S-GUS::GFP vector were transformed into LBA4404 Agrobacterium cells, then were injected to tobacco leaves according to the method of Zhao et al. [50]. The injected tobacco leaves were cultured in a greenhouse at 22 °C under a 16 h light/8 h dark cycle conditions for 72 h, and were punched for GUS staining, and GUS activity assay.

### 4.7. GUS Staining and Determination of GUS Activity

The perforated leaves were completely immersed in the GUS staining solution (Coolaber, Beijing, China), and vacuumed for 4 h. The leaves were placed in a shaker at 37 °C overnight, and finally decolorized with 75% ethanol to remove chlorophylls in plants. The criterion for complete decolorization was that the uninjected tobacco leaves appeared white. After complete decolorization, the perforated leaves were photographed under a super depth-of-field microscope to record the GUS staining. GUS activity was determined using Plant Glucosidase (GUS) ELISA Research Kit (MEIMIAN, Yancheng, China) according to the kit instructions.

### 4.8. Genetic Transformation and Identification of proMeSSIII-1 Promoter in Cassava

The pCAMBIA1304-proMeSSIII-1-GUS::GFP vector was transformed into SC8 cassava, and the genetic transformation process was referred to the methods of Wang et al. [51]. Total DNA of regenerated plants were used as a template to amplify the *gusA* gene on the pCAMBIA1304-proMeSSIII-1-GUS::GFP vector.

### 4.9. Hormone Treatments in proMeSSIII-1 Transgenic Cassava

The *proMeSSIII-1* transgenic cassava histocultured seedlings were treated with 100 μM ABA, 100 μM MeJA, and 100 μM ET (the hormones were dissolved in ethanol). Under the aseptic conditions, the stems of the *proMeSSIII-1* transgenic cassava seedlings were cut to several parts and cultured on MS medium at 28 °C under a 16 h light/8 h dark cycle at 50% relative humidity for 15 d. The newly growing cassava seedlings with uniform growth were selected for hormone spraying treatment and sampled after 0 h, 2, 6, 12, and 24 h of hormone treatments. The seedlings on MS medium without any treatment were used as control.

The tuber roots of *proMeSSIII-1* transgenic cassava grown in field for six months were cut into slices of about 3 mm in thickness. The slices were treated with hormones of 100 μM ABA, 100 μM MeJA, and 100 μM ET for 0, 2, 6, 12, and 24 h for sample collections according to the method of Yan et al. [52]. Slices without any treatment were used as controls. All samples were subjected to GUS staining and GUS activity assay. All experiments included at least three biological replicates per treatment.

### 4.10. Statistical Analysis

The data are presented as the mean ± SD and the data of three independent experiments were analyzed with a one-way analysis of variance. The value *p* ≤ 0.05 was considered significant with GraphPad Prism 8 software.

## Figures and Tables

**Figure 1 ijms-25-04711-f001:**
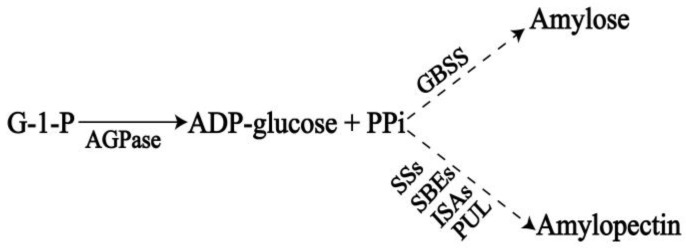
The pathway of starch synthesis.

**Figure 2 ijms-25-04711-f002:**
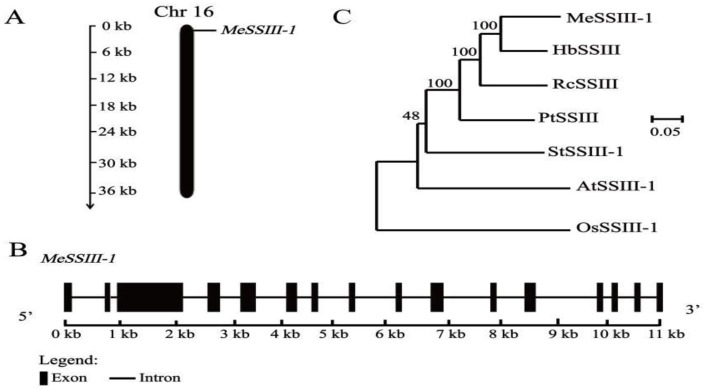
Chromosomic localization, gene structure of *MeSSIII-1* gene, and phylogenetic tree. (**A**) Chromosomic localization. (**B**) Exon–intron structure of the *MeSSIII-1* gene. Exons are shown as rectangular boxes, introns are shown as black lines. (**C**) Phylogenetic tree of MeSSIII-1 and SSIII proteins, in which HbSSIII from *H. brasiliensis*, RcSSIII from *R. communis*, PtSSIII from *P. trichocarpa*, StSSIII from *Solanum tuberosum*, AtSSIII-1 from *Arabidopsis thaliana*, and OsSIII-1 from *O. sativa* Japonica Group. The phylogenetic tree was constructed using MEGA 7.0 software and the neighbor-joining (NJ) method (1000 bootstraps).

**Figure 3 ijms-25-04711-f003:**
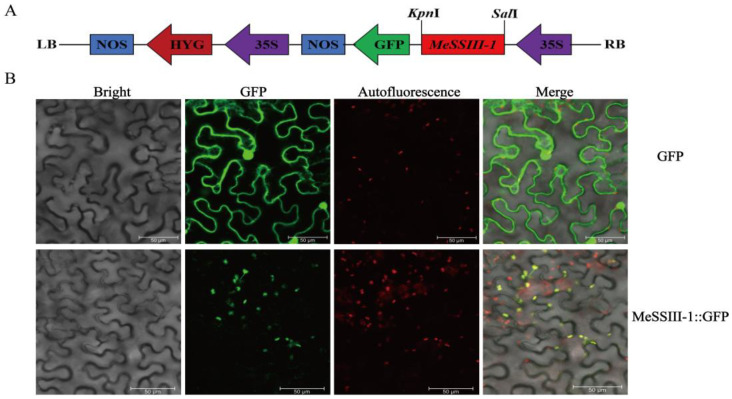
Subcellular localization of MeSSIII-1 protein in tobacco epidermal cell. (**A**).Schematic diagram of pCAMBIA1300-35S-MeSSIII-1::GFP vector construction; LB indicates the left border and RB indicates the right border of the pCAMBIA1300-35S-GFP vector. (**B**) The fluorescence signal detected in transfected cells. From left to right, the picture is a bright field, GFP fluorescence field, chlorophyll autofluorescence field, and merged field. Bar = 50 μm.

**Figure 4 ijms-25-04711-f004:**
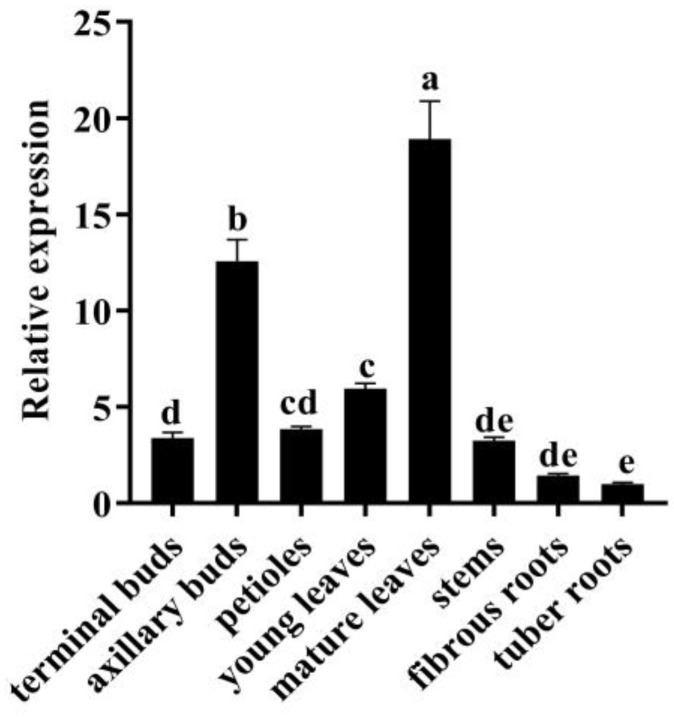
Expression analysis of *MeSSIII-1* gene in cassava tissues or organs. *MeTublin* were used as internal controls. The expression levels of the tuber roots were set to a value of 1. Data represent means of three biological repeats ± standard deviations. Different lowercase letters indicated significant differences (*p* < 0.05).

**Figure 5 ijms-25-04711-f005:**
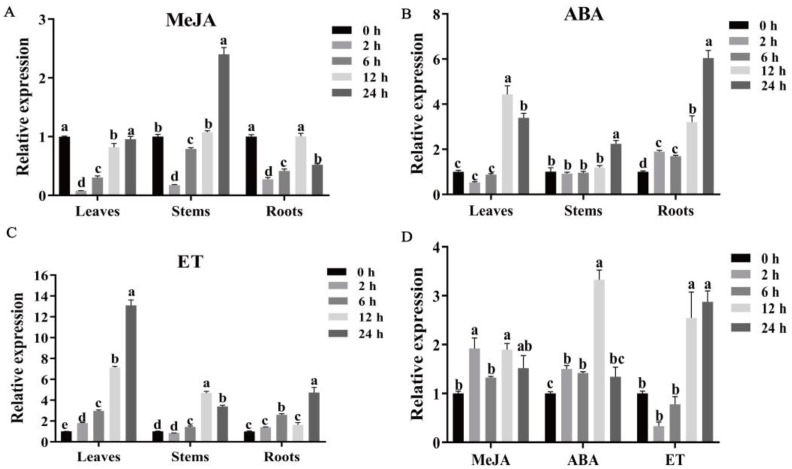
Expression patterns of *MeSSIII-1* under MeJA, ABA, and ET treatments in cassava histocultured seedlings and tuber roots. Cassava histocultured seedlings were treated with 100 μM MeJA (**A**), 100 μM ABA (**B**) and 100 μM ET (**C**). (**D**) Tuber roots were treated with 100 μM MeJA, 100 μM ABA, and 100 μM ET. *MeTublin* were used as internal controls. The expression levels of the appropriate controls were set to a value of 1. Data represent means of three biological repeats ± standard deviations. Each replicate was composed of six cassava histocultured seedlings. Different lowercase letters indicated significant differences (*p* < 0.05).

**Figure 6 ijms-25-04711-f006:**
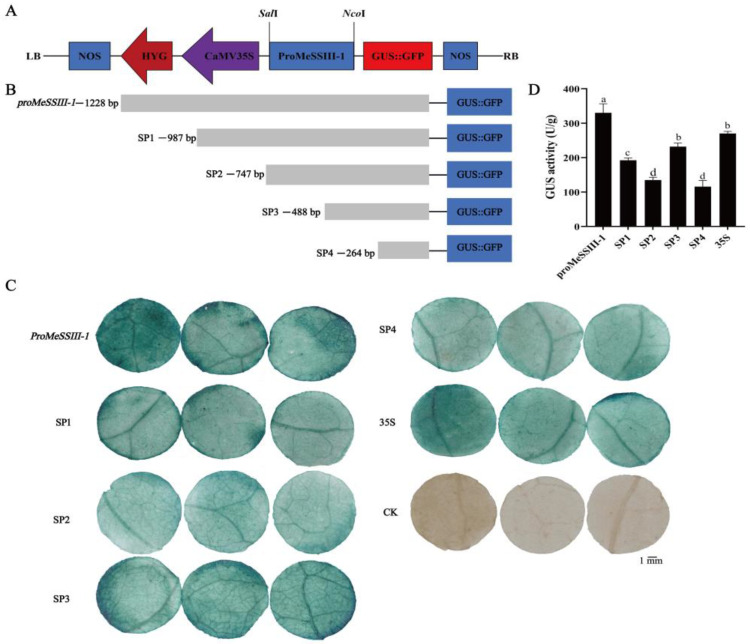
Analysis of the truncated *MeSSIII-1* promoter activity by 5’-deletions. (**A**) Schematic diagram of pCAMBIA1304-35S-proMeSSIII-1-GUS::GFP vector; LB indicates the left border and RB indicates the right border. (**B**) Schematic diagram of the truncated fragments of *proMeSSIII-1*. Five *MeSSIII-1* promoter fragments were constructed into pCAMBIA1304-35S-GUS::GFP to replace 35S promoter. (**C**) GUS staining of tobacco transiently transformed with different truncated *MeSSIII-1* promoters. (**D**) GUS activity of tobacco transiently transformed with the truncated *MeSSIII-1* promoters. CK indicated untreated tobacco leaves, which was used as negative control. 35S indicated tobacco leaves injected with empty vector pCAMBIA1304-35S-GUS::GFP, which was used as a positive control. Data represent means of three biological repeats ± standard deviations. Each replicate was composed of three tobacco. Different lowercase letters indicated significant differences (*p* < 0.05).

**Figure 7 ijms-25-04711-f007:**
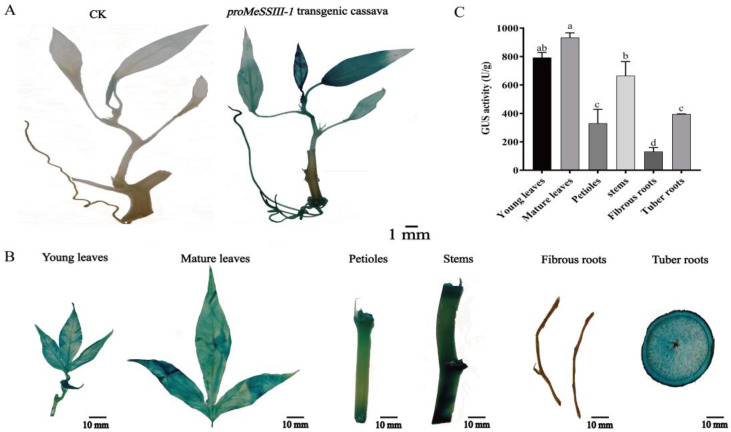
Tissue expression patterns of *proMeSSIII-1* in transgenic cassava. (**A**) Identification of *proMeSSIII-1* transgenic cassava by GUS staining. (**B**) Histochemical GUS staining in different tissues of *proMeSSIII-1* transgenic cassava. CK was a non-transgenic regeneration seedling. (**C**) GUS activity in different tissues of *proMeSSIII-1* transgenic cassava. Data represent means of three biological repeats ± standard deviations. Different lowercase letters indicated significant differences (*p* < 0.05).

**Figure 8 ijms-25-04711-f008:**
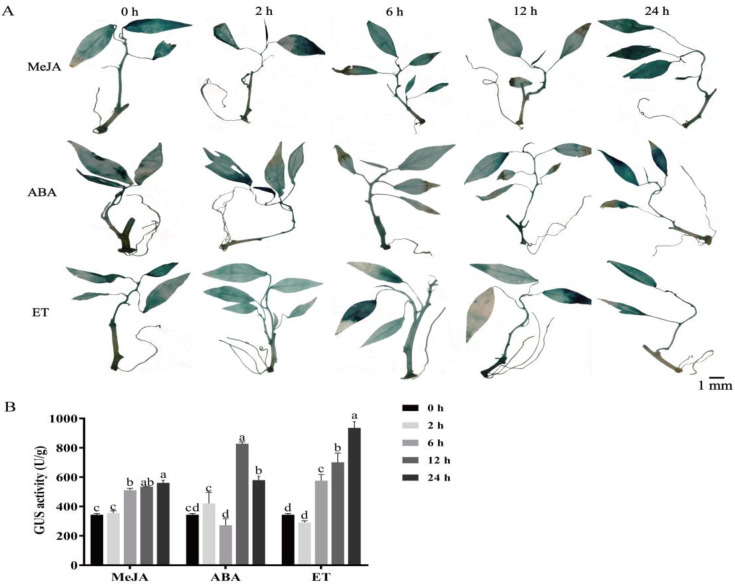
Analysis of *proMeSSIII-1* activity in histocultured seedlings under hormone treatments. (**A**) GUS staining of *proMeSSIII-1* transgenic cassava treated with ABA, MeJA, and ET at different time points. (**B**) GUS activity of *proMeSSIII-1* transgenic cassava treated with ABA, MeJA, and ET at different time points. Data represent means of three biological repeats ± standard deviations. Each replicate was composed of six cassava histocultured seedlings. Different lowercase letters indicated significant differences (*p* < 0.05).

**Figure 9 ijms-25-04711-f009:**
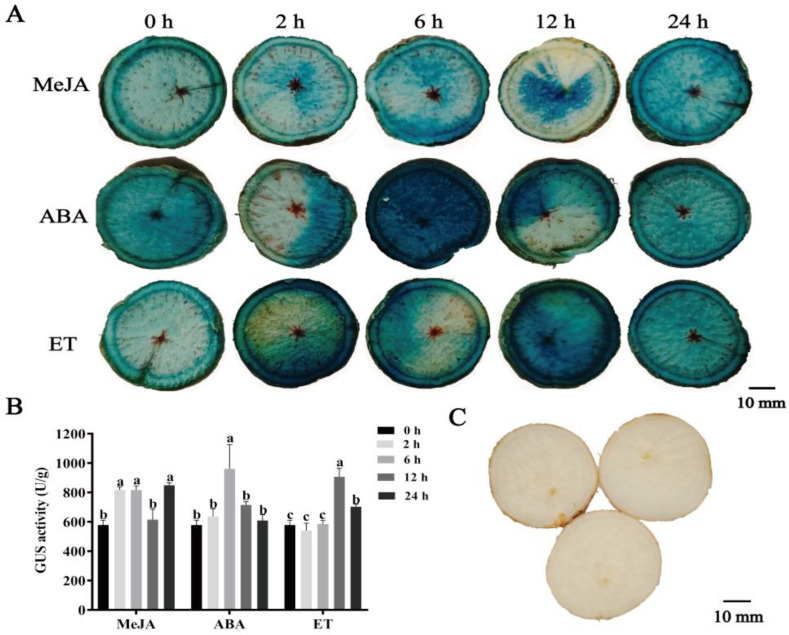
Analysis of *proMeSSIII-1* activity in tuber roots under hormone treatments. (**A**) GUS staining in *proMeSSIII-1* transgenic cassava tuber roots treated with ABA, MeJA, and ET. (**B**) GUS activity in *proMeSSIII-1* transgenic cassava tuber roots treated with ABA, MeJA, and ET. (**C**) GUS staining of root tubers from non-transgenic cassava. Data represent means of three biological repeats ± standard deviations. Different lowercase letters indicated significant differences (*p* < 0.05).

**Table 1 ijms-25-04711-t001:** Sequence, number, and function of *cis*-acting elements in *proMeSSIII-1*.

*Cis*-Elements	Number	Function	Core Sequence
ABRE	1	*cis*-acting element involved in the abscisic acid responsiveness	ACGTG
ERE	6	ethylene-responsive element	ATTT TAAA
CGTCA-motif	2	*cis*-acting regulatory element involved in the MeJA-responsiveness	CGTCA
TGACG-motif	2	*cis*-acting regulatory element involved in the MeJA-responsiveness	TGACG

## Data Availability

Data is contained within the article and Appendix A.

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
