# Peer review of "Functional Characterization of the MeSSIII-1 Gene and Its Promoter from Cassava"

_ijms, 2024, doi:10.3390/ijms25094711_

Round 1
Reviewer 1 Report
Comments and Suggestions for Authors
The article is devoted to the study of such an important process as starch synthesis, insofar as it concerns the operation of the cassava MeSSIII-1 gene encoding soluble starch synthase, the organization of its promoter and its interaction with a number of phytohormones. The data obtained in this study are undoubtedly of interest not only to scientists working with cassava but also to a wider range of readers and can be extrapolated to the processes of starch synthesis in other plant species. The work is mostly well thought out, implemented and written however there are a few remarks.
1. It is necessary to mention the full Latin name of cassava and the decoding of the designation of the cultivar used, at least in the Materials and Methods.
2. Caption for figure 1. Line 119. When first mentioning the Latin name of a species, it is necessary to write it in full and not in abbreviated form (S. Tuberosum, A. thaliana).
3. Captions for Figures 5 and 6 lack a decoding of the CK designation.
4. Typos – line 292 after link 33, apparently there should have been a comma? Line 399 “The” must begin with a lowercase letter.
5. There is not much discussion of the results obtained in the Discussion section. Lines 301-303, instead of assuming that the results of the study will provide a lot of useful information, it would be logical to present this information and discuss it. The study has already been carried out and in this section the results of the study and the contribution this study made to the understanding of the mechanisms of transcriptional regulation are discussed.
6. Conclusion section - it may be worth combining this with the Discussion section, since the Discussion section is not very long and just needs final conclusions. As it stands, the Conclusion section is a repetition of the main results; instead, I would like to see more of an interpretation of the results and possible prospects for further development of this study.
7. The Materials and Methods lack information about how the study described in section 2.3 of the Results was conducted.
8. Materials and methods – lines 395-396, there is no explanation why the gusA gene was amplified from the pCAMBIA-1304-35S-GUS:GFP construct, especially if this section (4.8) concerns the identification of the MeSSII-1 gene promoter.
Author Response
Response 1: I have added the full Latin name of cassava in line 77 and the designation of the cultivar used in line 358.
Response 2: I have added the full Latin name of S. Tuberosum and A. thaliana in figure 1, line 131.
Response 3: I have decoded the CK designation in the Figures 6 and 7, Line 226-227, Line 248.
Response 4:I have corrected these two errors. Line 311, Line 436.
Response 5: I have revised this part of the content according to your suggestions. Line 320-324.
Response 6: I have rewritten the conclusion section. Line 349-355. I hope to meet your requirements.
Response7: Agree. I have modified. Line 406-408.
Response8: We have corrected this error, which should be the gusA gene on the pCAMBIA1304-proMeSSIII-1-GUS::GFP vector. Line 441.
Reviewer 2 Report
Comments and Suggestions for Authors
The work “Functional Characterization of the MeSSIII-1 gene and its promoter from Cassava” by Xiao-Hua Lu and colleagues investigate the possibility that the expression of MeSSIII-1, which is important in the process of starch biosynthesis in Cassava, could be regulated by hormones. To this end, authors performed a series of experiments including qRT-PCR and the generation of GUS transcriptional reporter line to see not only whether overall treatment of hormones (ABA, MeJA and ET) could impact on MeSIII-1 expression but also to see in which organs this regulation occurs.
The introduction of the work is well written. “Material and Methods” section must be completed with the missing information. Results are overall consistent, even if they require a couple of corrections and completions.
A series of comment is listed below.
2.2. Subcellular Localization of MeSSIII-1 Protein
Line 126, 127 – The impression that the signal is in the cell membrane is due to the presence of vacuole pushing the cytoplasm in the periphery. Since the GFP was not constructed with a TAG to direct it in the plasma membrane, I exclude that it is in membrane. Moreover, in general, to claim that a protein is located in a specific compartment, it is advisable to show that its signal colocalize with the signal of a marker line specific for that specific compartment.
Fig.2 line 136, and MeM line 363, 364 – Green channel and red channel are too general description. Moreover, In MeM exact parameters used for acquisition should be reported (confocal type, excitation and emission lambda, filter, laser power, pinhole, objective)
2.3. Expression Patterns of MeSSIII-1 in Cassava Tissues
Line 140, 141 - the expression of MeSSIII-1 gene in mature leaves and axil-140 lary buds was dominantly obvious… I think that “dominantly obvious” is not a scientific sentence. Please remove.
Fig.3 and MeM – IMPORTANT, you should indicate the reference gene used to normalize the expression level of MeSIII-1. Also, the statistical test used must be indicated.
2.5. Expression Profiles of MeSSIII-1 Gene under Hormone Treatments
Line 169 and MeM – “histocultured seedlings”, please describe properly the procedure and the composition of the medium.
Fig.4 -Please indicate the statistical test that was used. Also, same as the comment in Fig.3.
2.6. Analysis of MeSSIII-1 Promoter Activity
Line 201 – “medicated”, please replace with “mediated”
Line 296-205 and MeM – From the description it is not clear whether the 35S is present in all the generated constructs. Was the 35S sequence removed when the various promoters were introduced?
2.7. Histochemical Localization of proMeSSIII-1 in Cassava
Line 218 – please specify that we are talking about “promoter activity”.
Line 228 – please replace “activates” with “is active”.
Line 230 – “These results hypothesize” … Please replace with something like “let us to hypothesize”.
Fig.5, 6, 7, and 8 – Indicate the number of replicates.
Line 240-242 – “The results showed that GUS staining and GUS activity had similar trends and showed a decrease first, then followed by an increase under treatment of the ABA and ET hormones”. I think that GUS staining and GUS activity do not allow to monitor the “decreasing”, GUS is too stable.
Material and Methods
4.1. Plant Materials and Growth Conditions
Please indicate the composition of the medium for histocultured seedlings. Also, I would expect that seeds were sterilezed. If is that the case, please indicate the sterilization process.
4.5. Expression Pattern of MeSSIII-1 Gene
Please describe how hormonal treatments were performed. Also, if ABA, ET, and MeJa were dissolved in a solvent like DMSO or ethanol, was the control performed using the same concentration of solvent? As a general comment, please describe what CK is at least the first time is cited (and in MeM).
Author Response
Response 1: Thank you for pointing this out. I have revised. Line 139.
Response 2: I have revised. Line 144, 406-409.
Response 3: Agree. I have modified. Line 147-148.
Response 4: Agree. I have modified. Line 151-153.
Response 5: I have added a description of the composition of histocultured seedlings and culture medium in section 4.1. Line 363-366.
Response 6: I have modified. Line 201-204.
Response 7: I have modified. Line 212.
Response 8: The four fragments and the full-length 1228 bp sequence were substituted for the 35S promoter within the pCAMBIA1304-35S-GUS::GFP vector, respectively. Line 208-210, 426-427.
Response 9: Yes, we are talking about “promoter activity”. I have modified in line 223-224.
Response 10: I have modified. Line 241.
Response 11: Agree. I have modified. Line 243.
Response 12: Thank you for pointing this out. I have revised in Fig.5, 6, 7, and 8.
Response 13: I have modified. Line 255-256.
Response14: Thank you for pointing this out. I have revised in 4.1. Line 363-366.
Response 15: Thank you for pointing this out, ABA, ET, and MeJa were dissolved in ethanol. and CK meaning in the text. Regarding the definition of CK, I have made separate annotations in the captions of Figure 6 and Figure 7. In section 2.5 (Figure 5), CK was was changed to 0 h.
Reviewer 3 Report
Comments and Suggestions for Authors
The study addresses the actual topic of starch biosynthesis in cassava, an important agricultural crop. The results obtained are of obvious interest. Below are some comments and suggestions regarding the manuscript text.
1. It is advisable to add a diagram of the starches biosynthesis pathway to the INTRODUCTION section. This would contribute to a better understanding of the article text.
2. Line 106. In the text "The CDS sequence of MeSSIII-1 is 3441 bp in length, which encodes 1146 amino acids, and locates on chromosome 16 (Figure 1A). The molecular weight and theoretical isoelectric point of MeSSIII-1 are 35 kDa and 8.54, respectively." The average molecular weight of amino acids is about 110, i.e. 35 kDa is about 340 amino acid residues, not 1146. Please clarify this.
3. Fig. 3. What was taken as the unit of expression?
4. Fig. 4. What is СK?
5. Section 2.6. Where did the 35S promoter come from? From the original plasmid pCAMBIA-1304-35S-GUS:GFP? It is necessary to indicate to Fig. 5 legend if this vector was used as a control.
6. Are you sure that the MeSSIII-1 promoter variants activities in tobacco plants correspond to their activities in cassava plants?
7. Section 2.7. Has the absence of endogenous GUS activity been checked in cassava seedlings?
8. Section 2.8. Did you analyze whole seedlings? Indicate this in the text.
9. Section 2.8. What is the point of this experiment? The activity of the MeSSIII-1 promoter has already been determined by RT-PCR, section 2.5.
10. Figure 8. It is advisable to add photos confirming the endogenous GUS activity absence in the tubers.
11. Line 313. What is LUC?
12. Line 317. The text contains "AGPpase". May be "AGPase"?
13. Line 331. What is "histocultured seedlings"?
14. Section 4.6. How were truncated promoter fragments obtained? Using PCR? Describe in more detail.
15. Section 4.9. How were plants treated with ET? Please provide the treatment protocol.
16. Section 4.10. What is the sample size used for statistical analysis?
Author Response
Response1: Thank you for pointing this out. I agree with this comment. Therefore, I have added a diagram of the starches biosynthesis pathway to the INTRODUCTION section.
Response2: Thank you for pointing this out. Due to my negligence, I made a mistake in this part of the results and have made revisions in the text. Line 118-119.
Response 3: The expression levels of the tuber roots were set to a value of 1. Line 152.
Response 4: In this figure, CK was changed to 0 h under hormone treatments. Line 196.
Response 5: Thank you for pointing this out. 35S promoter come from the original plasmid pCAMBIA1304-35S-GUS::GFP. I have already explained in Figure legend. Line 208-210.
Response 6: Validating the functionality of a gene's promoter in tobacco serves as a crucial approach for the rapid assessment of plant promoters. It would be more ideal to detect their activities in cassava plants. However, a method for transient expression to test promoter activity in cassava leaves has not yet been established.
Response7: CK was a non-transgenic regeneration seedling as a control and did not GUS activity.
Response 8: Thank you for pointing this out. I analyzed whole seedlings and indicated in the text. Line254.
Response 9: To ensure the reliability of the conclusions regarding the promoter's response to hormone treatment, this study utilized both RT-PCR and GUS activity assays for verification.
Response10: We have added a picture of GUS staining of root tubers from non-transgenic cassava in Figure 9 to demonstrate the endogenous GUS activity absence in the tubers. Other literature has shown that there is no GUS activity in cassava tuber roots (A promoter toolbox for tissue-specific expression supporting translational research in cassava (Manihot esculenta)).
Response11: LUC is a luciferase, a reporter gene.This sentence has been revised in Line 335.
Response12: We have corrected this error. Line 339.
Response13: SC8 cassava histocultured seedlings were kept in MS medium (supplemented with 2% sucrose and 0.8% agarose, pH 5.8, 121℃20min autoclave sterilization) at 28°C under 16 h/8 h (light/darkness) for 30 d.
Response14: Truncated promoter fragments were truncated into four segments by PCR. Line 425.
Response15: Thank you for pointing this out. I have provide the treatment protocol in Section 4.9.
Response16: I have modified. Line 464.
Reviewer 4 Report
Comments and Suggestions for Authors
Functional Characterization of the MeSSIII-1 gene and its promoter from Cassava
The abstract is qualified only in the last three lines (31-33). The others, including the background, the aims, the methods, and the results, are not well written.
1. In lines 20-22, the sentence should be written due to its ambiguousness and abundance of “starch synthesis” term.
2. In lines 20-23, the two sentences are not well connected. The second sentence does not contradict the first one. Therefore, the latter objectives of the current study are not clearly presented.
3. The methodology how the gene and its promoter were isolated is not clearly presented.
4. “the highest levels”, what are the actual values?
5. Avoid using pronouns to replace terms. This could reduce the readability of the manuscript.
6. “might”? If this is unsure, please not report this in the abstract.
The introduction should go through a major revision.
1. “vascular plants” should be used to replace “higher plants”
2. The sentence in lines 37-38 should be cited.
3. The sentence in lines 38-41 should be separated into three smaller sentences. This is a typical example for a run-on sentence. Please apply the same to other run-on sentences.
4. The cassava introduction should be a separate paragraph.
5. The sentence in lines 94-96 should be cited.
6. The details in lines 98-102 should not appear in the introduction, instead, they should be placed in the materials and methods.
The results should be slightly modified.
1. Materials and methods should not be mentioned in the results.
2. There are also many run-on sentences in this section.
Discussion should be proofread again.
1. In lines 304-306, the sentence should be cited.
2. There are many typos in this section. Please check this section again.
3. Still, there are many run-on sentences.
Materials and methods
1. Where is the thermal cycle of the PCR?
2. Because it was unable to access to the Supplementary data 1 to see if origin of the primers is well presented or not, I just recommended to mention their origin in the manuscript.
Conclusions are acceptable.
Comments on the Quality of English LanguageMinor editing of English language required
Author Response
Response1: I have rewritten this part, hoping to meet your requirements.Line22-23.
Response2:I have rewritten these sentences. Line 22-23.
Response3: Thank you for pointing this out. I agree with this comment. Therefore, I have clearly presented the separation of genes and promoters in sections 4.3 of materials and methods.
Response4:I have rewritten this part. Line 26-27.
Response5: Agree,I have revised pronouns from the abstract. Line 26-28.
Response6: Agree,I have removed “might”. Line 32.
Response7: Agree,I have revised. Line 59.
Response8: Agree, I have cited references in lines 59.
Response9: I have revised these sentences. Line 59-62.
Response10: I have revised this part accroding to your suggestion. Line 77-885.
Response11: Thank you for pointing this out. The expression levels of seven soluble starch synthesis genes in different tissues and organs as well as tuber root development periods were calculated according to the SC8 cassava RNA sequencing data (unpublished). Line 109-110.
Response12: I have revised this part accroding to your suggestion. Line 110-112.
Response13: Agree, I have revised. Line 156, 212, 234.
Response14: I have revised accroding to your suggestion.
Response15: the sentence have cited In lines 304-306.
Response16: I have corrected these error. Line 300,311, 321-325,335.
Response17: Agree,I have revised.
Response18: Thank you for pointing this out. I agree with this comment. Therefore, I have described the PCR program in sections 4.3 and 4.5 of materials and methods.
Response19: Thank you for pointing this out. I agree with this comment .I have supplemented the sources of primers in the methods and attachments.
Reviewer 5 Report
Comments and Suggestions for Authors
I have finished my reading of the manuscript entitled "Functional Characterization of the MeSSIII-1 gene and its promoter from Cassava" and am satisfied that the overall presentation of the manuscript is good with very professional style of writing. The manuscript provides the information about the MeSSIII-1 gene and its promoter in the Cassava plants which significantly adds to the knowledge of starch synthesis in the plants and in specific tissues there-off. However, there are some points that I have raised that can be addressed by the authors to improve the overall quality of the manuscript
1. In line 22 authors are talking about only amylopectin or complete starch.
2. MeJA and ABA are two very different categories of hormones with very different functions physiologically, what made you use them together?
3. It will be more appropriate to start the introduction by mentioning promoters and genes involved in the synthesis of starch. 3rd paragraph can be made 1st with more emphasis on promoters followed by other paragraphs.
4. Why was cassava chosen? It would be informative to mention a few lines about it.
5. In lines 74-84, section on gene expression needs more elaboration considering the theme of the article.
6. Result is very well written, with good figure quality and there is not much need to change anything in the section with only one question in section 2.2 - why only tobacco was considered for this experiment?
7. Lines 279-281 can be rewritten
8. can you please add one or two references in the line 303 in support of your findings.
9. In sections 4.4, 4.6 and 4.9 some reference can be added for the technique used.
10. Conclusion should be rewritten to better justify the results obtained, as the present text feels more like summary of results.
Author Response
Response1: I have revised. Line 22.
Response2: Abscisic acid (ABA) response element (ABRE), MeJA response elements CGTCA-motif and TGACG-motif were both found on the proMeSSIII-1 promoter sequence (Table 1).
Response3: Agree,I have revised the introduction accroding to your suggestion.
Response4: The third paragraph of introduction described the importance of cassava.
Response 5: Thank you for pointing this out. I agree with this comment. Therefore,I have added some elaboration on gene expression. Line 45-51.
Response6: The method of transient expression of promoters in tobacco is relatively fast.
Response 7: Thank you for pointing this out.,I have revised.
Response8: Agree,I have added references in the line 303.
Response9: Reference were added for the technique used in sections 4.4, 4.6 and 4.9.
Response10: The conclusion has been rewritten. Line 348-354.